Delimiting the genera of the Ficinia Clade (Cypereae, Cyperaceae) based on molecular phylogenetic data

http://orcid.org/0000-0002-0763-0780 Muasya A. Muthama 1 2 muthama.muasya@uct.ac.za
http://orcid.org/0000-0003-0285-722X Larridon Isabel 2 3
1 Department of Biological Sciences, Bolus Herbarium, University of Cape Town , Rondebosch, Cape Town , South Africa
2 Identification and Naming, Royal Botanic Gardens Kew , Richmond, Surrey , UK
3 Department of Biology, Systematic and Evolutionary Botany Lab, Ghent University , Gent , Belgium
Sosa Victoria
Electronic publication date: 2021 Jan 26
Publication date: 2021
Volume: 9
Electronic Location ID: e10737
Received 2020 Aug 20; Accepted 2020 Dec 18
Copyright: © 2021 Muasya and Larridon
Copyright year: 2021
Copyright holder: Muasya and Larridon
License: This is an open access article distributed under the terms of the Creative Commons Attribution License, which permits unrestricted use, distribution, reproduction and adaptation in any medium and for any purpose provided that it is properly attributed. For attribution, the original author(s), title, publication source (PeerJ) and either DOI or URL of the article must be cited.
License URL: https://creativecommons.org/licenses/by/4.0/

Keywords: Taxonomy, Phylogenetics, Cyperaceae, Classification, Nomenclature, Cape Flora, Isolepis, Ficinia

Funding: University of Cape Town and the South African National Research Fund (NRF) This work was supported by the University of Cape Town and the South African National Research Fund (NRF, incentive fund for rated researchers). The funders had no role in study design, data collection and analysis, decision to publish, or preparation of the manuscript.

==============================
Generic delimitations in the Ficinia Clade of tribe Cypereae are revisited. In particular, we aim to establish the placement of annual species currently included in Isolepis of which the phylogenetic position is uncertain. Phylogenetic inference is based on two nuclear markers (ETS, ITS) and five plastid markers (the genes matK, ndhF, rbcL and rps16, the trnL intron and trnL-F spacer) data, analyzed using model based methods. Topologies based on nuclear and plastid data show incongruence at the backbone. Therefore, the results are presented separately. The monophyly of the smaller genera (Afroscirpoides, Dracoscirpoides, Erioscirpus, Hellmuthia, Scirpoides) is confirmed. However, Isolepis is paraphyletic as Ficinia is retrieved as one of its clades. Furthermore, Ficinia is paraphyletic if I. marginata and allies are excluded. We take a pragmatic approach based on the nuclear topology, driven by a desire to minimize taxonomic changes, to recircumscribe Ficinia to include the annual Isolepis species characterized by cartilaginous glumes and formally include all the Isolepis species inferred outside the core Isolepis clade. Consequently, the circumscription of Isolepis is narrowed to encompass only those species retrieved as part of the core Isolepis clade. Five new combinations are made (Ficinia neocapensis, Ficinia hemiuncialis, Ficinia incomtula, Ficinia leucoloma, Ficinia minuta). We present nomenclatural summary at genus level, identification keys and diagnostic features.

Introduction

The paradigm shift towards recognition of genera as monophyletic entities has necessitated changes in generic circumscription (Humphreys & Linder, 2009). Within Cyperaceae, a number of changes have been made within the last decade, for example the merger of segregate genera into the paraphyletic core in Carex L. (Global Carex Group (GCG), 2015) and Cyperus L. (Larridon et al., 2011a, 2011b, 2013, 2014; Bauters et al., 2014). A number of genera have been found to be polyphyletic, especially in the tribe Schoeneae, resolved by reclassification of entities and naming of a number of lineages as new genera (Elliott & Muasya, 2017; Larridon et al., 2018; Larridon, Verboom & Muasya, 2018; Barrett, Wilson & Bruhl, 2019). Each of the four Cyperaceae genera recognized by Linnaeus (1753) has been reclassified over the years, with Linnaeus’ circumscription of Scirpus L. as encompassing species with bisexual flowers and spiral glume arrangement representing the most heterogenous assemblage. Embryo morphology data (Van der Veken, 1965; Goetghebeur, 1986; Semmouri et al., 2019) have unequivocally demonstrated that lineages with distinct morphology were included in Scirpus. In his seminal treatment of the family, Goetghebeur (1998) placed the 24 taxa previously named Scirpus by Linnaeus (1753) in the tribes Cypereae, Fuireneae and Scirpeae, with S. sylvaticus L. being the only species described by Linnaeus recognized as a true Scirpus and two of the species recognized as members of Isolepis R.Br.

Within tribe Cypereae, the Ficinia Clade (sensu Muasya et al., 2009b) comprises taxa whose placement has been most contentious. Goetghebeur (1998) diagnosed tribe Cypereae to include species characterised by either a Cyperus or a Ficinia type embryo, where glumes are arranged distichously (Cyperus and allies) or spirally (Isolepis, Ficinia Schrad., Scirpoides Ség). Lineages bearing perianth segments were added to the clade based on molecular phylogenetic data, moving Hellmuthia Steud. from Chysitricheae (Vrijdaghs et al., 2006; Muasya et al., 2009a, 2009b), Erioscirpus Palla from Scirpeae (Yano et al., 2012), and recognizing southern African taxa previously placed in Scirpus as a distinct genus Dracoscirpoides Muasya (Muasya et al., 2012). Furthermore, the delimitation of Scirpoides has been altered to exclude Afroscirpoides García-Madr. and Muasya (García-Madrid et al., 2015) and the addition of two species that were ambiguously placed (Browning & Gordon-Gray, 2011; Reid et al., 2017). These genera are annual to perennial herbs, have basal leaves which vary in blade development, have considerable variation in inflorescence and floral morphology, and are diagnosed by a combination of morphological features (see Table 2 in García-Madrid et al., 2015).

Generic delimitation between Isolepis and Ficinia is based on few morphological characters. Isolepis have a varied habit (annual to perennial) and are widespread, whereas Ficinia are perennial and predominantly occur within the Cape flora and in Africa (Goetghebeur, 1998; Muasya & Simpson, 2002). A further distinction is the presence of a gynophore in Ficinia, but several species having a gynophore and occurring outside Africa were previously excluded from the genus. For example, the New Zealand iconic sand dune taxon (Desmoschoenus spiralis (A.Rich.) Hook.f.) is embedded within core Ficinia (as Ficinia spiralis (A.Rich.) Muasya and De Lange; Muasya & De Lange, 2010), and the more widespread Ficinia nodosa (Rottb.) Goetgh., Muasya and D. A. Simpson was retained in Isolepis despite having a well developed gynophore (Muasya, Simpson & Goetghebeur, 2000). Furthermore, Isolepis may not be monophyletic as Ficinia is one of the three to four clades recovered in Isolepis (Muasya & De Lange, 2010; García-Madrid et al., 2015; Hinchliff & Roalson, 2013; Spalink et al., 2016; Semmouri et al., 2019). Challenges on distinguishing Ficinia from Isolepis have persisted over the last 200 years, as evident from at least one in six of the currently recognized Ficinia species having a validly published epithet in Isolepis (Govaerts et al., 2020).

We use an expanded molecular phylogenetic study to investigate the generic limits in the Ficinia Clade. We infer the phylogenetic relationships and placement of ambiguously placed Isolepis species, namely (1) I. hemiuncialis (C. B. Clarke) J. Raynal and I. incomtula Nees—which have been previously recovered as sister to the core Isolepis/Ficinia clade; (2) I. marginata (Thunb.) A. Dietr. and allies (I. antarctica (L.) Roem. and Schult., I. capensis Muasya, I. leucoloma (Nees) C. Archer, I. minuta (Turrill) J. Raynal)—previously recovered as sister to core clade of Ficinia. The aim is to establish whether the genera in the Ficinia Clade, particularly Isolepis and Ficinia, are monophyletic, and to evaluate what characters diagnose the inferred (sub)clades.

Materials and Methods

Ethics statement

Part of the specimens studied were collected during field expeditions predominantly in Western Cape province of South Africa funded by grants from the National Research Foundation and with additional support from the University of Cape Town. Permit to collect these specimens were issued by the Cape Nature authorities (CN35-28-5831). The other specimens studied are available in publicly accessible herbaria (BOL, K; B. Thiers, continuously updated, http://sweetgum.nybg.org/science/ih/) and voucher details provided in Tables S1 and S2.

Nomenclature and taxonomy

A nomenclatural study including the taxonomic history of the genus and its species, critical for the correct coining of the new names and the proper use of prior ones, was performed. The electronic version of this article in Portable Document Format will represent a published work according to the International Code of Nomenclature for algae, fungi and plants, and hence the new names contained in the electronic version are effectively published under that Code from the electronic edition alone. In addition, new names contained in this work which have been issued with identifiers by IPNI will eventually be made available to the Global Names Index. The IPNI LSIDs can be resolved and the associated information viewed through any standard web browser by appending the LSID contained in this publication to the prefix “http://ipni.org/”. The online version of this work is archived and available from the following digital repositories: PeerJ, PubMed Central and CLOCKSS.

Molecular study

The Ficinia Clade, our focus group, includes the genera Afroscirpoides (one species), Dracoscripoides (three species), Erioscirpus (two species), Ficinia (81 species), Hellmuthia (one species), Isolepis (75 species) and Scirpoides (four species). A total of 166 ingroup accessions were sequenced (Table S1), representing: one species of Afroscirpoides (100%), three Dracoscripoides (100%), one Erioscirpus (50%), 64 (plus three undescribed) Ficinia (78%), one Hellmuthia (100%), 57 Isolepis (plus some infraspecific taxa) (76%) and three Scirpoides (75%). The outgroup taxa, selected based on Semmouri et al. (2019) and Larridon et al. (2021), consists of 114 accession (Table S2) representing the six genera of tribe Fuireneae and the genus Cyperus, the only genus of the Cyperus Clade of tribe Cypereae after Androtrichum was recently synonymised with Cyperus (Pereira-Silva et al., 2020). The DNA extraction protocol, primers, and material and methods for PCR amplification and sequencing follow Viljoen et al., 2013.

Sequences were assembled and edited in Geneious R8 (http://www.geneious.com, Kearse et al., 2012), aligned using MAFFT 7 (Katoh, Asimenos & Toh, 2009; Katoh & Standley, 2013) with “maxiterate” and “tree rebuilding number” set to 100 (long run), afterwards, alignments were checked manually in PhyDE 0.9971 (Müller et al., 2010).

We first inferred the gene trees for each of the regions separately to identify potential incongruence. As there were no instances of conflict at well-supported nodes between the two nuclear markers, and between the five chloroplast makers, the matrices of the regions were concatenated into a nuclear dataset (Data S1) including ETS and ITS, and in a chloroplast dataset (Data S2) including the genes matK, ndhF, rbcL and rps16, the trnL intron and trnL-F spacer for the downstream analyses. PartitionFinder 2.1.1 (Lanfear et al., 2012) was used to determine an appropriate data-partitioning scheme from potential partitions that were defined a priori (in this case, each marker was treated as a separate partition), as well as the best-fitting model of molecular evolution for each partition, using the Bayesian Information Criterion. For the nuclear dataset, PartitionFinder confirmed the a priori data-partitioning scheme, and the GTR+I+Γ (invgamma) model of sequence evolution was determined to be the best-fitting model for the two nrDNA markers. For the chloroplast dataset, PartitionFinder suggested partitioning the data in four partitions (matK+rps16, ndhF, rbcL, the trnL intron and trnL-F spacer) and the GTR+Γ (gamma) model of sequence evolution was determined to be the best-fitting model for all partitions except for rbcL for which the GTR+I+Γ (invgamma) model was suggested.

Maximum likelihood (ML) analyses of the optimally partitioned data were performed using RAxML 8.2.10 (Stamatakis, 2014). The search for an optimal ML tree was combined with a rapid bootstrap analysis of 1,000 replicates. Additionally, partitioned analyses were conducted using Bayesian Inference in MrBayes 3.2.6 (Ronquist et al., 2012). Rate heterogeneity, base frequencies, and substitution rates across partitions were unlinked. The analysis was allowed to run for 100 million generations across two independent runs with four chains each, sampling every 10,000 generations. Convergence, associated likelihood values, effective sample size values and burn-in values of the different runs were verified with Tracer 1.5 (Rambaut & Drummond, 2007). The first 25% of the trees from all runs were excluded as burn-in before making a majority-rule consensus of the posterior distribution trees using the “sumt” function. All phylogenetic analyses were run using the CIPRES portal (http://www.phylo.org/; Miller, Pfeiffer & Schwartz, 2011), and were executed for both full and reduced sampling alignments. Trees were drawn using TreeGraph2 (Stöver & Müller, 2010).

Morphological study

Herbarium specimens of BOL, BR, GENT, K, NBG, PRE (B. Thiers, continuously updated, http://sweetgum.nybg.org/science/ih/) were studied morphologically using a Leica (Leica Microsystems, Wetzlar, Germany) binocular microscope. Measurements were made with a ruler and digital calipers (e.g., leaf and culm length), or using a binocular microscope with graticule (e.g., spikelet and glume length). When examining width, this was done near the middle of the organ (e.g., middle of the nutlet).

Results

Molecular study

Analyses of the individual markers show congruence within the nuclear and plastid markers, as well as congruence among the methods of analyses (Fig. S1). However, the nuclear (Fig. 1A) and plastid phylogenies (Fig. 1B) had conflicting backbone topologies and are therefore presented separately. Regardless, there is strong support in both data sets for the tribe Cypereae with the Cyperus and Ficinia clades as sister (Figs. S1–S4). In the plastid topology (Fig. 1B), Afroscirpoides diverged first, then strongly supported sister clades comprising (i) Erioscirpus sister to Scirpoides, and (ii) a clade comprising Hellmuthia, Dracoscirpoides, Isolepis and Ficinia. In the nuclear topology, there is a grade showing successive divergence starting with Erioscirpus, Afroscirpoides, Dracoscirpoides, Scirpoides, Hellmuthia, then Isolepis and Ficinia. In both analyses, there is strong support for the monophyly of the Dracoscirpoides, Hellmuthia and Scirpoides, but Isolepis is paraphyletic with Ficinia as one of the clades.

Figure 1 The majority-rule consensus Bayesian Inference of the Ficinia Clade, showing posterior probabilities at nodes, based on MrBayes analyses.

(A) Concatenated nuclear, (B) concatenated plastid DNA sequence data.

The position of Isolepis hemiuncialis and I. incomtula differed in the two analyses. These two species formed an early diverging grade leading to a polytyomy among core Isolepis in the plastid topology (Fig. 1B), but were part of the I. marginata clade in the nuclear phylogeny (Figs. 1 and 2). On the other hand, I. marginata and allied species (I. antarctica, I. capensis, I. leucoloma and I. minuta) were consistently resolved as part of a clade including Ficinia eligulata and sister to the core Ficinia clade. The nuclear topology is better resolved, showing subclades in core Isolepis which coincide with current infrageneric groups, but these groups are not clearly discrenable in the plastid topology. In addition, the nuclear ITS alignment shows a three nucleotide insertion (ATA; position 1890–1892, Data S1), unique to the core Isolepis clade and lacking in the outgroup as well other Isolepis (I. hemiuncialis, I. incomtula, I. marginata and allies) and Ficinia.

Figure 2 A simplified majority-rule consensus Bayesian Inference of the ingroup based on the concatenated nuclear DNA sequence data.

Morphological study

Table 1 summarises the morphological diversity among genera in the ingroup. All Ficinia Clade taxa share the presence of Cyperus or a modified type (Ficinia) embryo. They are annuals or perennials; are mostly scapose, though multiple nodes are observed in Ficinia (e.g., F. trichodes (Schrad.) B. D. Jacks.); and have leaf blades well developed or reduced to a lobe, with or without ligule. The inflorescence is diverse (single terminal, for example, I. ludwigii (Steud.) Kunth, Fig. 3A; capitate, for example, F. ecklonea (Steud.) Nees, Fig. 3I; pseudolateral, for example, Afroscirpoides; to anthelate, for example, S. burkei (C. B. Clarke) Goetgh., Muasya and D. A. Simpson). Glume arrangement is predominantly spiral, with distichous arrangement in some species of Isolepis and Ficinia (e.g., I. levynsiana Muasya and D. A. Simpson, Fig. 3B; F. distans C. B. Clarke). The flowers are bisexual and occurring in most florets, but dioecy is observed in Afroscirpoides. While majority of taxa lack perianth, these occur in Erioscirpus (large and plumose as seen in Eriophorum L.), Dracoscirpoides (bristles) and Hellmuthia (scales). Nutlets are trigonous to oval in cross section, with the base extended to form a gynophore in the majority of Ficinia.

Table 1 Comparison of genera in the Ficinia clade, reflecting the revised classification.

	Erioscirpus Palla	Afroscirpoides García-Madr. & Muasya	Dracoscirpoides Muasya	Scirpoides Ség.	Hellmuthia Steud.	Isolepis R.Br.	Ficinia Schrad.	
Life form	Perennial	Perennial	Perennial	Perennial	Perennial	Annual and perennial	Perennial, few annual	
Leaf blade	Well developed	Poorly developed (<5 mm long)	Well developed	Mostly poorly developed	Mostly poorly developed	Mostly well developed	Mostly well developed	
Inflorescence type	Anthelate	Capitate, pseudolateral	Capitate, pseudolateral	Capitate, anthelate, pseudolateral	Capitate	Capitate, terminal	Capitate, spike, pseudolateral	
Glume arrangement	Spiral	Spiral	Spiral	Spiral	Spiral	Spiral, few distichous	Spiral, few distichous	
Perianth type in fertile flowers	>6 Bristles, cotton-like	Absent	6 (7) Bristles, scabrid	Absent	3 Scales in lower flowers	Absent	Absent (single case recorded)	
Gynophore	Absent	Absent	Absent	Absent	Absent	Absent	Mostly present	
Embryo type	Cyperus	Cyperus	Cyperus	Cyperus	Cyperus	Cyperus and Ficinia	Ficinia	
Number of species	2	1	3	4	1	70	~90	
Distribution	Asia	Southern Africa	Southern Africa	Southern Africa, Eurasia, Americas	Southern Africa (Cape)	Southern and Tropical Africa, Australasia, Europe, Americas,	Southern and Tropical Africa, Australasia, circumpolar	

Figure 3 Morphological diversity in the Ficinia Clade.

(A) I. ludwigii (Schlechter 1821 BOL); (B) I. levynsiana (Muasya 2274 BOL); (C) I. hemiuncialis (Muasya et al. 6040 BOL); (D) I. incomtula (BOL6370); (E) I. leucoloma (Levyns 7618 BOL); (F–G) I. marginata (Verboom 1241 BOL); (G) I. marginata (in situ); (I) F. ecklonea (Muasya and Stirton 3215 BOL). Images of the inflorescences were made using a Leica S9i and habit photograph (H) made by Muthama A. Muasya.

Species in the Isolepis and Ficinia clades vary in subtle morphological and ecological features (Table 1; Fig. 3). Isolepis hemiuncialis and I. incomtula are annual species, whose gross morphology and ecology is similar to annual species in the core Isolepis clade and I. marginata (and allies). Ficinia is unique in its ecology, frequently growing as perennial in dry habitats and unlike perennial Isolepis species that are restricted to wetlands. An additional feature, the presence of a gynophore is unique to Ficinia, eventhough several species (e.g., F. filiformis (Lam.) Schrad., F. trollii (Kük.) Muasya and D. A. Simpson) lack this feature and vestigial gynophore occasionally occur in Isolepis marginata. Glume texture is chartaceous in Isolepis (including I. hemiuncialis and I. incomtula), whereas it is cartilaginous in Ficinia (including I. marginata and allies).

Discussion

This study has inferred the phylogeny of the Ficinia Clade using a large species sample (78% of species) and Sanger sequencing of nuclear and plastid markers. The patterns observed are similar to previous studies (Muasya et al., 2009a, 2009b; Muasya & De Lange, 2010; Hinchliff & Roalson, 2013; Spalink et al., 2016; Semmouri et al., 2019), confirming the monophyly of the smaller genera but recovering Ficinia to be nested in Isolepis. The backbone differs between the plastid and nuclear topology, especially relating to the position of Dracoscirpoides which is positioned between Afroscirpoides and Scirpoides (nuclear, Fig. 1A) or between Hellmuthia and the Isolepis/Ficinia clade (plastid, Fig. 1B). Similarly, these varied topologies have been observed in previous studies (García-Madrid et al., 2015). As the majority of the deeper nodes in the Ficinia Clade are highly supported (PP above 0.95, Figs. 1A and 1B), we suspect that the observed pattern is caused by evolutionary phenomena such as reticulate evolution (Pelser et al., 2010).

The phylogenetic position of Isolepis hemiuncialis and I. incomtula is unstable, shifting based on the markers analyzed. These taxa form a lineage (or grade) separate from the core Isolepis and Ficinia clades based on plastid data in this and previous studies (García-Madrid et al., 2015; Spalink et al., 2016), and a similar pattern was observed in a combined plastid and nuclear analysis (rps16 and ITS; Muasya & De Lange, 2010). In addition, the nuclear markers differ in their placement of these samples, with ITS having a pattern similar to plastid markers (similarly observed in Figs. S1 and S2 of García-Madrid et al. (2015)), but ETS placing these species as part of clade with I. marginata (similarly observed in Figs. S3 and S4 of García-Madrid et al. (2015)). In the combined nuclear matrix, these two species are part of the I. marginata clade. In contrast, I. marginata (and allied species) have been consistently observed to be forming a clade sister to core Ficinia in separate and combined analyses, and these species are in same clade with F. eligulata Gordon-Gray ex Muasya from the Drakensberg Mountain.

A unique three base-pair insertion in ITS2 further supports the uniqueness of the core Isolepis. This insertion is missing in I. hemiuncialis, I. incomtula as well as the species in the I. marginata clade, and can therefore be used as a synapomorphy for the core Isolepis clade. Similar use of indels, located at the 5.8S gene of the nuclear ribosomal DNA, as synapomorphies has been suggested for the Cypereae (Yano et al., 2012) and Cyperaceae (Starr et al., 2007).

A number of the genera in the ingroup can be distinguished unambiguously based on one or few characters (Table 1). The presence and type perianth segments, even though perhaps arising independently, are unique in Dracoscirpoides (scabrid bristles; Muasya et al., 2012), Erioscirpus (cotton-like bristles; Yano et al., 2012) and in Hellmuthia (scale-like; Vrijdaghs et al., 2006). Among the taxa lacking perianth segments, Afroscirpoides and Scirpoides have densely tufted culms which have reduced leaf blades (>5 mm, but some Scirpoides have well developed leaf blades), with the former having dioecious individuals whereas the later has bisexual florets. Ficinia is most similar in gross morphology and ecology to Afroscirpoides and Scirpoides, diagnosed by the presence of a cupular disk (gynophore; Vrijdaghs et al., 2005) at the base of the nutlets (except in several species where the trait is lost; Muasya et al., 2014). Isolepis is most similar to Ficinia, sharing presence of bisexual florets and glumes with well defined parallel veins, but differing in Isolepis lacking the gynophore. The glume texture appears to offer additional separation, being chartaceous to hyaline (herbaceous; Muasya & Simpson, 2002) in Isolepis but cartilaginous (or coriaceious) in Ficinia.

Generic boundaries within Isolepis and Ficinia have been noted as problematic. Eleogiton, still recognized as distinct in some floras (e.g., Germany, Kadereit et al., 2016) based on possessing multiple internodes and peduncle termination in a single terminal spikelet, is confirmed to be a clade in Isolepis (subgenus Fluitantes; Muasya et al., 2001; Muasya & Simpson, 2002). In Ficinia, Sickmania Nees has been previously recognized based on a capitate inflorescence with multiple leaf-like bracts (F. radiata) whereas Desmoschoenus has primary bracts adnate to axis and covering congested spikelets (Goetghebeur, 1998). The phylogenetic inference showing I. maginata and other annual species that lack a gynophore being closer to Ficinia further blurs the generic boundaries.

Cyperaceae has experienced shifting generic classification in the last two decades. The paradigm shift to recognize monophyletic genera (Humphreys & Linder, 2009) accompanied by the use of DNA sequence data have enabled disentangling phylogenetic relatedness of taxa obscured by extreme morphological modification. Several highly diversified lineages appear to have been split into genera based on one of few characters, at times such characters arising independently. This phenomenon was epitomized Cyperus, now recognized as a single genus (Larridon et al., 2011a, 2011b, 2013, 2014; Bauters et al., 2014), where 13 segregate genera were diagnosed based on morphology of reproductive structures (spikelet size and organization, nutlet orientation, style branching; Muasya et al., 2009b). This study supports a further refinement within the Cypereae, recognizing the core Isolepis and an enlarged Ficinia at generic level.

We speculate that the Ficinia clade evolved in southern Africa, given that majority of lineages and species occur in the region. Diversification in Isolepis and Ficinia has occurred since the Miocene (Besnard et al., 2009), perhaps ecologically driven by aridification associated with onset of the Mediterranean climate (Linder & Verboom, 2015), where emerging traits include annual life form, colonization of permanently wet habitats, sprouting regeneration driven by the frequent fires in sclerophyllous habitats, and ant dispersal of seeds (gynophore in Ficinia; Bond & Slingsby, 1983). Within southern Africa, the Ficinia Clade members are predominantly occurring in the Greater Cape Flora and exhibit the typical diversification pattern whereby lineages in the Fynbos are older than those in the Succulent Karoo biome (Verboom et al., 2009). Dispersal out of the Cape appears to be predominantly to other similar habitats, especially in Mediterranean Eurasia (Erioscirpus, Isolepis, Scirpoides), within temperate zones of high mountains in tropical Africa (Dracoscirpoides, Ficinia, Isolepis, Scirpoides) and austral temperate areas (Ficinia, Isolepis). Dispersal to Australasia in Isolepis has been accompanied by hybridization in Isolepis (Ito et al., 2016).

Taxonomic treatment

The current generic classification is supported for the smaller genera (Afroscirpoides, Dracoscirpoides, Erioscirpus, Hellmuthia, Scirpoides). However, Isolepis is paraphyletic as Ficinia is one of its clades as well as Ficinia is paraphyletic if I. marginata and allies are excluded. We acknowledge the conflicting topology between the nuclear and plastid phylogenies, particularly regarding the position of I. hemiuncialis and I. incomtula, opting to follow the nuclear phylogeny. We take a phragmatic approach, to recognize clades that will minimize nomenclatural changes, by adopting a classification framework based on the nuclear phylogeny (Fig. 2). We therefore recognize an expanded concept of Ficinia, to include annual species with mostly cartilaginous glumes and lacking a gynophore (occasionally a gynophore is observed among Australian I. marginata, see Fig. 3F). As a consequence, Isolepis is now considered in a narrower concept which encompasis the core Isolepis and excludes the seven annual species placed within the Ficinia clade (I. antactica, I. capensis, I. hemiuncialis, I. incomtula, I. leucoloma, I. marginata and I. minuta). Subclades within Isolepis can be recognized as infrageneric groups, recognizing four subgenera where three are similar to classification by Muasya & Simpson (2002) but elevating sect. Proliferae to subgeneric rank. In Ficinia, previous infrageneric groups (Clarke, 1897; Pfeiffer, 1921) are not supported, but the two clades each with subclades could form basis for future infrageneric classification. Formal taxonomic changes are made here, but we note the need of a comprehensive taxonomic revision of Ficinia.

Key to the species of Ficinia clade genera

1 Plants perennial or annual; perianth segements absent4

1 Plants perennial; perianth segements present2

2 Perianth segment cotton-like, restricted to Asia1. Erioscirpus

2 Perianth segment bristle or scale-like; restricted to southern Africa3

3 Slender plants, culm < 4 mm diameter; perianth bristle-like; in Drakensberg and surrounding areas2. Dracoscirpoides

3 Robust plant, culm > 5 mm diameter; perianth scale-like; in Cape area3. Hellmuthia

4 Perennial habit; inflorescence in globose clusters of over 10 spikelets; nutlet lacking a gynophore; embryo Cyperus-type5.

4 Perennial or annual habit; inflorescence in clusters mostly of under 10 spikelets; nutlet with or without a gynophore; embryo Cyperus- or Ficinia-type6.

5 Plants leafless; inflorescence pseudolateral, dioecious4. Afroscirpoides.

5 Plants leafless or well developed blades; inflorescence with bisexual flowers, pseudolateral to anthelate5. Scirpoides.

6 Glumes chartaceous, nutlet lacking gynophore6. Isolepis.

6 Glumes cartilaginous, nutlet mostly bearing gynophore7. Ficinia

1. Erioscirpus Palla, Bot. Zeitung (Berlin) 54: 151 (1896). Type species – Erioscirpus comosus (Wall.) Palla, designated here.

Two species of perennial hemicryptophytes, diagnosed on presence of cotton-like perianth. Taxonomic revision as part of regional flora, for example, Flora of Pakistan (http://www.tropicos.org/Project/Pakistan).

Distributed in Asia, from Iran to China, occurring in shallow soil and rocky crevices, at 700–2,300 m.

2. Dracoscirpoides Muasya, S. African J. Bot. 78: 108 (2012). Type species—Dracoscirpoides falsa (C.B.Clarke) Muasya.

Three species of perennial hemicryptophytes or rhizomatous geophytes, taxonomy revised in Muasya et al. (2012).

Restricted to southern Africa, occurring in montane grasslands.

3. Hellmuthia Steud., Syn. Pl. Glumac. 2: 90 (1855). Type species—Hellmuthia membranacea (Thunb.) R. W. Haines and Lye.

Monotypic, hemicryptophytes or rhizomatous geophytes, diagnosed by presence of scale-like perianth. Taxonomic studies in local flora (Archer & Muasya, 2012).

Restricted to South Africa, occurring coastal areas in calcareous sandy soils in the Cape Flora.

4. Afroscirpoides García-Madr. & Muasya, Taxon 64: 698 (2015). Type species—Afroscirpoides dioeca (Kunth) García-Madr.

Monotypic, densely tufted hemicryptophytes or rhizomatous geophytes, diagnosed by dioecious flowers borne in dense globose inflorescences whose bract terminates in a sharp-pointed tip.

Restricted to southern Africa, occurring in seepages and streambeds in brackish habitats.

5. Scirpoides Ség., Pl. Veron. 3: 73 (1754). Type species—Scirpoides holoschoenus (L.) Soják, designated here.

Four species recognized in Govaerts et al. (2020), but two additional species segregated from the widespread S. holoschoenus by García-Madrid et al. (2015).

Widespread in Mediterannean habitat in Mexico, Canary Is. through northern Africa and Eurasia to W. Himalaya, South Africa.

6. Isolepis R.Br., Prodr. Fl. Nov. Holland.: 221 (1810). Type species—Isolepis setacea (L.) R.Br.

About 70 species recognized here, after moving seven species to Ficinia. Nearly a third of species are therophytes, rest are hemicryptophytes or rhizomatous geophytes. Most recent and comprehensive taxonomic revision in Muasya & Simpson (2002).

Nearly cosmopolitan distribution, with highest species densities in austral temperate southern Africa and Australasia.

7. Ficinia Schrad., Commentat. Soc. Regiae Sci. Gott. Recent. 7: 143 (1832). Type species—Ficinia gracilis Schrad.

About 90 species are recognized here, including the annual species transferred from Isolepis. Majority of species are perennial hemicryptophytes or rhizomatous geophytes, adapted to survive frequest fires in the Fynbos biomes, but also few annual and pyrophytic short-lived perrenials. The most comprehensive taxonomic study of Ficinia was part of the Flora Capensis (Clarke, 1897) and recent synopsis of the Cape Flora (Archer & Muasya, 2012). Ongoing studies reveal existence of undescribed species and the Ficinia is among the highest priority Cypeaceae for taxonomic revision in southern Africa.

Predominantly occurring in southern Africa in the Cape Flora and extending into montane areas of tropical Africa. Two species occur in Australasia, among which F. nodosa is nearly circumpolar.

Species transferred from Isolepis to Ficinia in this study:

The annual Isolepis species forming a clade sister to F. eligulata (Fig. 2) are here transferred into Ficinia.

Ficinia neocapensis Muasya, nom. nov.

Isolepis capensis Muasya, Kew Bull. 57: 305 (2002). (basionym)

Ficinia hemiuncialis (C. B. Clarke) Muasya, comb. nov.

Scirpus hemiuncialis C. B. Clarke in É. A. J. De Wildeman, Pl. Nov. Horti Then. 1: 23 (1904). (basionym)

Isolepis hemiuncialis (C. B. Clarke) J. Raynal, Adansonia, n.s., 17: 55 (1977).

Ficinia incomtula (Nees) Muasya, comb. nov.

Isolepis incomtula Nees, Linnaea 10: 154 (1835). (basionym)

Ficinia leucoloma (Nees) Muasya, comb. nov.

Cyperus leucoloma Nees, Linnaea 10: 133 (1835). (basionym)

Isolepis leucoloma (Nees) C. Archer, Bothalia 28: 42 (1998).

Ficinia marginata (Thunb.) Fourc., Trans. Roy. Soc. South Africa 21: 76 (1932).

Scirpus marginatus Thunb., Prodr. Pl. Cap.: 17 (1794). (basionym).

Isolepis marginata (Thunb.) A. Dietr., Sp. Pl. 2: 110 (1833).

There appears to be continuity in the number of spikelets per inflorescence, with materials at extreme ends recognized as I. marginata versus I. antarctica. We retain the two taxa as a single species, Ficinia marginata, and refrain from making a new combination pending a detailed taxonomic study of the complex.

Ficinia minuta (Turrill) Muasya, comb. nov.

Scirpus minutus Turrill, Bull. Misc. Inform. Kew 1925: 69 (1925). (basionym)

Isolepis minuta (Turrill) J. Raynal, Adansonia, n.s., 17: 56 (1977)

Conclusions

This study aimed to establish the phylogenetic position of contentious annual species currently placed in Isolepis and to test the monophyly of the genera. All the other smaller genera in the Ficinia clade (Afroscirpoides, Dracoscirpoides, Erioscirpus, Hellmuthia and Scirpoides) are monophyletic. There is unambiguous placement of I. marginata and allies (I. antarctica, I. capensis, I. leucoloma and I. minuta) as a clade within Ficinia and not part of the core Isolepis clade. Inclusion of I. hemiuncialis and I. incomtula into a clade including I. marginata is supported by the nuclear phylogeny, but these taxa are placed in a grade outside the core Isolepis clade. We propose the reclassification of these Isolepis species, resolved outside the core Isolepis, as species within Ficinia. The proposed classification will add taxa lacking the gynophore, the currently used diagnostic character for Ficinia, with the core Isolepis diagnosed by a combination of morphology (e.g., chartaceous glumes, no gynophore) and a unique indel in ITS.

Supplemental Information

Supplemental Information 1 Alignment of the concatenated nuclear DNA sequence data.

Click here for additional data file.

Supplemental Information 2 Alignment of the concatenated plastid DNA sequence data.

Click here for additional data file.

Supplemental Information 3 Voucher information of the Ficinia clade accession used in the molecular study, including GenBank accession numbers.

GenBank accessions with prefix ‘MW’ are newly reported in this study.

Click here for additional data file.

Supplemental Information 4 Voucher information of the outgroup accessions used in the molecular study, including GenBank accession numbers.

Click here for additional data file.

Supplemental Information 5 The majority-rule consensus Bayesian Inference of all studied taxa, showing posterior probabilities at nodes, based on MrBayes analyses of the concatenated nuclear DNA sequence data.

Click here for additional data file.

Supplemental Information 6 The majority-rule consensus Bayesian Inference of all studied taxa, showing posterior probabilities at nodes, based on MrBayes analyses of the concatenated plastid DNA sequence data.

Click here for additional data file.

Supplemental Information 7 The optimum Maximum Likelihood tree topology of all studied taxa, showing bootstrap support values at nodes, based on RAxML analyses of the concatenated nuclear DNA sequence data.

Click here for additional data file.

Supplemental Information 8 The optimum Maximum Likelihood tree topology of all studied taxa, showing bootstrap support values at nodes, based on RAxML analyses of the concatenated plastid DNA sequence data.

Click here for additional data file.

We thank J.-A. Viljoen, R. Skelton and S. Wiswedel for assistance with laboratory work.

Additional Information and Declarations

Competing Interests

Author Contributions

Field Study Permissions

DNA Deposition

Data Availability

New Species Registration

Isabel Larridon is an Academic Editor for PeerJ.

A. Muthama Muasya conceived and designed the experiments, performed the experiments, analyzed the data, prepared figures and/or tables, authored or reviewed drafts of the paper, and approved the final draft.

Isabel Larridon analyzed the data, prepared figures and/or tables, authored or reviewed drafts of the paper, and approved the final draft.

The following information was supplied relating to field study approvals (i.e., approving body and any reference numbers):

Permit to collect these specimens were issued by the Cape Nature authorities (CN35-28-5831).

The following information was supplied regarding the deposition of DNA sequences:

Voucher information of all material used in the molecular study, including GenBank accession numbers, are available in the Supplemental Files.

The following information was supplied regarding data availability:

Alignment of the concatenated nuclear/plastid DNA sequence data are available in the Supplemental Files.

The following information was supplied regarding the registration of a newly described species:

Ficinia neocapensis Muasya, nom. nov.: 77213741-1

Ficinia hemiuncialis (C. B. Clarke) Muasya, comb. nov.: 77213737-1

Ficinia incomtula (Nees) Muasya, comb. nov.: 77213738-1

Ficinia leucoloma (Nees) Muasya, comb. nov.: 77213739-1

Ficinia minuta (Turrill) Muasya, comb. nov.: 77213740-1.

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
