# Peer review of "Delimiting the genera of the Ficinia Clade (Cypereae, Cyperaceae) based on molecular phylogenetic data"

_PeerJ, doi:10.7717/peerj.10737_

## Round 0.1 · original submission · Minor Revisions

Both reviewers as well as myself coincided with the decision of minor revisions, one of them attached a file including his comments made on the text. Please consider these suggestions, mainly in the references and figure legends.

·

Basic reporting

The article has a clear structure, is well written and provides enough background information. The quality of the used methods, interpretation of the data and other content is very high.

Figures and tables are clear and relevant to the content.

In some parts of the texts such as the 'Discussion' chapter should be carefully checked for grammar, such as for lacking articles. I marked some of these with sticky notes is the PDF.

Experimental design

This study forms the next stage in a continuous effort of the authors to understand the molecular phylogeny of the Ficinia clade of Cypereae and its implication on the taxonomy of this challenging group. This study provides further essential basis for a modern generic classification of Cypereae, and provides the foundation to tackle a final big challenge on this group: a modern monograph of the genus Ficinia. It is not the first molecular study performed on this group and most findings coroborate previous studies. However the use of a large array of markers from both nuclear and chloroplast DNA, state of the art analysis methods and a translation of the results to taxonomical changes within the group of study, make this article worth publishing in PeerJ.

The used methods (choice of markers, statistical analysis, …) are well chosen and correctly applied. As all sequence data are or will be available from Genebank, this study is replicable.

Validity of the findings

The authors use a critical language. Results are well analyzed and conclusions are well stated. Underlying data have been provided.

Additional comments

I have a few suggestions and questions which could enhance the quality of this article, however I my opinion these are not essential to the acceptance of this article for publication:

An additional plate with (field) photographs showing key morphological & ecological characteristics of typical core Isolepis vs core Ficinia (e.g. chartaceous vs coriaceous glumes) and of the species newly transferred to Ficinia would be valuable.

Isolepis species either have a Cyperus or a Ficinia embryo type. As the Ficinia type is derived from the Cyperus type, the latter is expected to be present only in the most basal clades of Isolepis. Is there support for a unique origin of the Ficinia type embryo? Did the authors check the occurrence of the embryo type with the different clades found in the nuclear vs chloroplast analysis? Especially for Isolepis hemiuncialis, incomtula, … of which the position is currently still unclear, this might add information? In addition I suppose the taxa newly transferred to Ficinia all have a Ficinia embryo type?

·

Basic reporting

The reporting in this paper is very clear.

The introduction starts with a very comprehensive description of the taxonomical history of the Ficinia clade, followed by the challenges on
distinguishing Ficinia from Isolepis and the dificulties with the delimiations. A very clear aim is stated:
"The aim is to establish whether the genera in the Ficinia Clade, particularly Isolepis and Ficinia, are monophyletic, and to evaluate what characters diagnose the inferred (sub)clades."

With the material and method section gives enough detail to be able to reconstruct this research.

Result are clearly reported, easy to follow with the figures by hand.

The discussion handles the 'research question' / aim of this paper.

After reading this article, the "research question" is fully answered. This paper is very well structured: after reading this paper only once I could fully grasp the outcomes and understand the results. The text is not too long and easy to comprehend, which makes it very accessible for everyone interested in this subject!

The taxonomic treatment with a key to the genera is very useful.

The figures are good. Although, I would summarize the cut-out part of the 1A and 1B trees as 'Cyperus-clade' (or outgroup), as done in Fig. 2. Also, I would annotate Cyperus clade and Ficinia-clade on the trees?

Some small inconsistencies were found in the references, I annotated these in the pdf.

Experimental design

There is a well defined research question & aim in this paper: 'Investigating the generic limits of the Ficinia-clade'. The aim is to establish whether the genera in the Ficinia Clade, particularly Isolepis and Ficinia, are monophyletic, and to evaluate what characters diagnose the inferred (sub)clades.

This is very relevant. In the last decade there have been many changes within the Cyperaceae family and this paper handles one of the still unresolved issues.

With the material and method section gives enough detail to be able to reconstruct this research. I am happy to see that nuclear and plastid markers are not concatenated into one matrix but analysed separately. The large species sample makes this research very relevant!

Validity of the findings

I encourage the pragmatic approach that was chosen in adopting a classification framework based on the nuclear phylogeny (and thus recognizing clades that minimize nomenclatural changes).

The data on which the results and conclusions are based are available.

The conclusion is very well stated and there is a clear link to the original research question. Conclusion are based on the supported results.

---

## Round 0.2 · accepted · Accept

I am glad that in this article you tried to minimize nomenclatural changes in this group of Cyperaceae and that you found diagnostic characters. My only recommendation is that during the editorial process you change to words numbers below ten. In this version, you use numbers (e.g. in Discussion). Thank you for considering previous suggestions by reviewers.